# Stylized Handwriting Generation of Arbitrary Structures and OOV Expressions: A Decoupled Approach via Layout-Offsets

## Abstract

To truly understand human handwriting, machines must not only recognize glyphs but also generate them. However, most existing approaches are limited to synthesizing isolated characters or handwritten texts of linear sequences, whereas the stylized synthesis of handwriting with arbitrary layout structures remains largely underexplored, such as handwritten mathematical expression generation (HMEG). Existing approaches have failed to address such cases, as it is challenging to simultaneously generate complex layout structures and imitate calligraphic styles, especially for out-of-vocabulary (OOV) expressions. Inspired by how humans write, where layout structuring and glyph shaping are inherently separated, we therefore propose a glyph-layout decoupled paradigm for stylized HMEG. To better facilitate the generation of arbitrary layout structures, we leverage printed layouts as strong prior guidance and propose generating layout offsets instead of absolute positions. To achieve stylized glyph-layout synthesis, we further incorporate implicit context adaptation via cross-attention to jointly mimic structured layouts and calligraphic glyphs from reference examples. By treating reference layouts and glyphs as external implicit contexts, our model selectively attends to relevant stylistic features of each symbol and its bounding box. Experiments demonstrate that our method outperforms previous SoTA approaches in terms of visual quality, semantic and structural correctness, and style consistency for stylized HMEG.

## 1 Introduction

Handwriting plays a critical role in human education, creativity, and cultural preservation; thus, enabling machines to understand human handwriting is highly meaningful. However, an intelligent system should not be limited to handwriting recognition alone, but also possess the ability to generate human-like writing. With the recent advances in artificial intelligence techniques, it has witnessed significant progress in handwriting recognition (Guan et al., 2024; Castro et al., 2024; Guo et al., 2025) and generation (Ren et al., 2025; Pippi et al., 2025; Dai et al., 2023; Pippi et al., 2023a). Nevertheless, most existing approaches are confined to synthesizing isolated glyphs or linear handwritten texts, while handwriting synthesis with arbitrary layout structures remains largely underexplored, such as arbitrary handwritten mathematical expression generation (HMEG).

Therefore, this paper aims to generate stylized handwriting of arbitrary layout structures and out-of-vocabulary (OOV) expressions (e.g., stylized HMEG), a direction that remains rarely explored. In particular, we demonstrate that stylized HMEG is significantly more challenging than conventional handwritten text generation (HTG) of linear sequences, due to the following aspects: *(1) Arbitrary Spatial Structures:* In contrast to handwritten texts of linear sequences, handwritten mathematical expressions (HMEs) involve two-dimensional spatial structures that arrange intricate symbols and encode semantic relationships; *(2) Structured OOV Expressions:* Due to their structural complexity, mathematical expressions are more difficult to represent; incorporating OOV textual contents further amplifies its complexity and variability; *(3) Glyph-Layout Imitation:* Stylized HMEG requires simultaneously imitating both complex layout structures and calligraphic styles of glyphs, thus presenting greater challenges than HTG with linear structures.

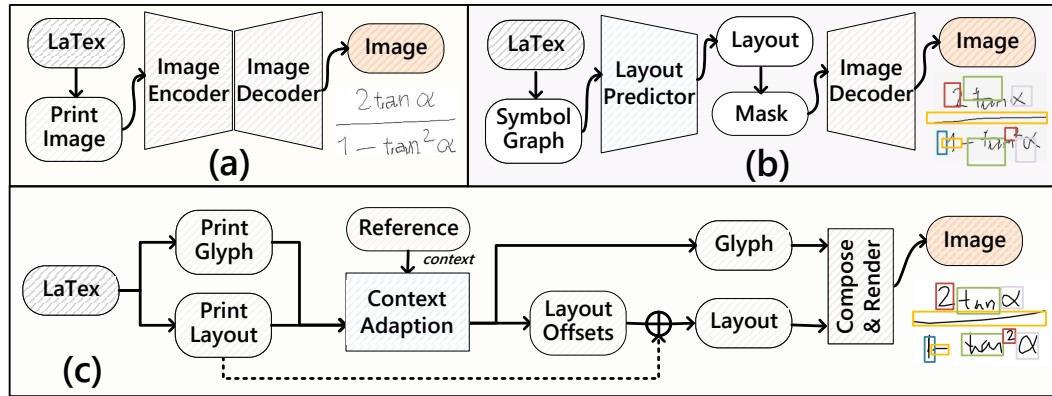

Figure 1: Comparison between our method and existing approaches for HMEG. **(a) Image-to-Image Translation** often synthesizes unrealistic and low-quality handwriting. **(b) Graph-to-Image Translation** frequently generates structurally incorrect or visually implausible HMEs, especially for OOV expressions. **(c) Our Layout-Glyph Decoupled Paradigm via Layout Offsets** can generate stylized handwriting of arbitrary layout structures and OOV expressions.

Although few recent works have attempted to explore HMEG, the aforementioned challenges remain largely unsolved. As illustrated in Fig. 1 (a) and (b), existing approaches can be broadly classified into two categories: *(1) Image-to-Image (I2I) Translation:* FormulaGAN (Springstein et al., 2021) and ControlNet (Zhang et al., 2023) addressed HMEG as I2I translation, which directly converts printed glyphs into stylized handwriting. However, it struggles to capture complex layout structures and diverse calligraphic styles of HMEs, often resulting in unrealistic and low-quality handwriting. *(2) Graph-to-Image (G2I) Translation:* Sg2Im (Johnson et al., 2018) and SgHMEG (Chen et al., 2024) proposed a G2I translation pipeline, which first generates layout heatmaps conditioned on LaTex symbol graphs and then decodes them into HMEs through an image decoder. Nevertheless, it remains empirically challenging for such cases to generate diverse and accurate layouts from symbol graphs (particularly for OOV expressions), which frequently leads to structurally incorrect or visually implausible HMEs. This is because directly generating layouts of unknown positions but with precise semantics is significantly difficult. More critically, ***none of the existing approaches attempted to tackle stylized HMEG***, which requires imitating both complex layout structures and calligraphic styles of glyphs conditioned on reference examples.

| Method | Estimate Layouts | OOV Gen. | Stylized Gen. | Quality↓ | | |
|---|---|---|---|---|---|---|
| | | | | LSD† | HWD | FID |
| CycleGAN | × | × | × | × | 0.688 | 84.14 |
| FomulaGAN | × | × | × | × | 0.717 | 74.68 |
| ControlNet | × | × | ✓ | × | 1.011 | 131.99 |
| Sg2Im | ✓× | × | × | 8.158 | 0.452 | 10.02 |
| SgHMEG | ✓× | × | × | 7.081 | 0.439 | 10.98 |
| Ours | ✓ | ✓ | ✓ | **5.271** | **0.151** | **5.60** |

Table 1: Feature-by-feature comparison of different models for HMEG. "✓×": generated layouts are inconsistent with HME images, "LSD": layout structure distance/similarity, "HWD": handwriting distance/score, †: only applicable to methods that generate layouts, "↓": lower values are better.

To address these challenges, we propose ***a decoupled baseline that leverages layout offsets*** for stylized handwriting generation of arbitrary structures and OOV expressions, as shown in Fig. 1 (c). Our approach is motivated by the following core ideas: *(1) Considering that human handwriting can be naturally decomposed into structured layouts and individual glyphs, as also suggested in Ren et al. (2025), perhaps a glyph-layout decoupled paradigm is more feasible for HMEG.* This is because directly generating stylized handwriting in such cases is highly challenging due to glyphs' arbitrary spatial arrangements, complex semantic relationships, and distinct calligraphic styles, especially for

OOV HMEs. *(2) Rather than directly predicting layouts of absolute positions, we leverage printed layouts as strong prior guidance and propose generating layout offsets instead.* This significantly simplifies the layout generation and empirically improves the semantic correctness, thus enabling handwriting generation of arbitrary structures. *(3) To achieve stylized HMEG, we incorporate Implicit Context Adaptation via cross-attention to jointly mimic the structured layouts and calligraphic glyphs from reference examples.* Reference layouts and glyphs serve as external implicit contexts, enabling cross-attention to selectively focus on relevant style features of each symbol and bounding box, thereby facilitating realistic and coherent handwriting generation.

Table 1 shows a feature-by-feature comparison of different models for HMEG. And our contributions are listed as follows:

- We propose a glyph-layout decoupled baseline via layout offsets for stylized HMEG, which can synthesize stylized handwriting of arbitrary layout structures and OOV expressions. Such a task remains largely underexplored, and our decoupled paradigm is demonstrated to be more feasible for stylized HMEG than conventional I2I or G2I approaches.

- We leverage printed layouts as strong prior guidance and propose generating layout offsets (instead of absolute layout positions). The strong prior guidance of template layouts largely facilitates the layout generation of arbitrary structures, especially for OOV expressions.

- To achieve stylized glyph-layout synthesis, we incorporate implicit context adaptation via cross-attention to jointly mimic structured layouts and calligraphic glyphs from references. By treating reference layouts and glyphs as external implicit contexts, it selectively attends to the relevant stylistic features of each symbol and its bounding box.

- Experiments demonstrate that our method outperforms previous SOTA approaches for stylized HMEG in terms of visual quality, structural and semantic correctness, and style consistency.

## 2 RELATED WORK

### 2.1 HANDWRITTEN TEXT SYNTHESIS WITH LINEAR SEQUENCES

HTG primarily aims to generate handwritten texts with linear structures. Primary approaches typically leveraged GANs (Alonso et al., 2019) to synthesize handwritten texts conditioned on specific contents, such as ScrabbleGAN (Fogel et al., 2020). Stylized HTG has further been investigated by integrating I2I translation techniques, including GANwriting (Kang et al., 2020), SLOGAN (Luo et al., 2022), VATr (Pippi et al., 2023a), HiGAN+ (Gan et al., 2022), etc. Due to their strong modeling and generative capabilities, diffusion models (Ho et al., 2020; Nichol & Dhariwal, 2021) have also emerged as powerful generative models for HTG, such as DiffusionPen (Nikolaidou et al., 2024), GC-DM (Ding et al., 2023), ZPL-LDM (Mayr et al., 2024), WordStylist (Nikolaidou et al., 2023), etc. Recent works have further emphasized the importance of layouts for stylized HTG, such as DLGOC (Ren et al., 2025). However, existing methods are limited to HTG of linear sequences.

### 2.2 HME GENERATION WITH COMPLEX STRUCTURES

Tremendous efforts have been made for HME recognition (Zhang et al., 2017; Li et al., 2022; Guo et al., 2025), while stylized HMEG of arbitrary layout structures remains largely underexplored. Few attempts have emerged recently, but all have failed to generate realistic HMEs with precise layouts and semantics, especially for OOV expressions. FormulaGAN (Zhu et al., 2017) leveraged I2I translation to directly transfer printed glyphs, often producing unrealistic and low-quality HMEs. Sg2Im (Johnson et al., 2018) and SgHMEG (Chen et al., 2024) introduced symbol graphs to generate layouts and then employed an I2I decoder to produce HMEs. However, such approaches typically produce structurally incorrect or visually implausible HMEs, particularly for unseen expressions. Nevertheless, stylized HMEG remains challenging as it requires simultaneously modeling complex layouts and calligraphic glyphs of reference examples.

## 3 METHODOLOGY

### 3.1 GLYPH-LAYOUT DECOUPLED GENERATION PARADIGM

**Problem Statement.** We aim to achieve stylized handwriting synthesis of arbitrary structures and OOV expressions. Given the reference example $\mathbf{X}_r \leftrightarrow (\mathbf{L}_r, \mathbf{G}_r)$ with the layout $\mathbf{L}_r = [b_1, \cdots, b_L]$ and its handwritten glyphs $\mathbf{G}_r = [g_1, \cdots, g_M]$, our objective is to generate target handwriting $\mathbf{X}_t \leftrightarrow (\mathbf{L}_t, \mathbf{G}_t)$ of the specific textual content $\mathcal{T} = [c_1, \cdots, c_N]$ while imitating both the structured layouts and calligraphic styles of $\mathbf{X}_r$, i.e.,

$$\mathbf{X}_t = \mathcal{G}(\mathbf{X}_r, \mathcal{T}) \ \Rightarrow \ (\mathbf{L}_t, \mathbf{G}_t) = \mathcal{G}(\mathbf{L}_r, \mathbf{G}_r, \mathcal{T}), \tag{1}$$

where $\mathcal{G}$ is the handwriting generator, $b_l \in \mathbf{L}$ is the bounding-box in layout $\mathbf{L}$, $c_t \in \mathcal{T}$ is the character symbol, and $g_m \in \mathbf{G}$ is a handwritten glyph of the specific character.

**Layout-Glyph Decoupled Generation Paradigm.** For stylized HMEG, it maximizes the likelihood of $\mathbf{X}_t$ conditioning on the reference $\mathbf{X}_r$ and the specific textual expression $\mathcal{T}$ as

$$P(\mathbf{X}_t | \mathbf{X}_r, \mathcal{T}) = P(\mathbf{L}_t, \mathbf{G}_t | \mathbf{L}_r, \mathbf{G}_r, \mathcal{T}). \tag{2}$$

Nevertheless, directly modeling the joint distribution of complex layouts and handwritten glyphs is overwhelmingly challenging, especially for HMEs of complex layouts and structured expressions. *Motivated by the observation that humans often possess an implicit awareness of the spatial layout of HMEs before producing individual glyphs, we hypothesize that layout planning and symbol generation can be decoupled.* Specifically, perhaps a more feasible solution is to decouple the layout and glyph generation processes, i.e., first separately generating the layout and glyphs, and then combining them to form the final HME. Thus, we can simplify and reformulate the problem as

$$P(\mathbf{L}_t, \mathbf{G}_t | \mathbf{L}_r, \mathbf{G}_r, \mathcal{T}) \ \Rightarrow \ P(\mathbf{L}_t | \cancel{\mathbf{G}_t}, \mathbf{L}_r, \cancel{\mathbf{G}_r}, \mathcal{T}) P(\mathbf{G}_t | \cancel{\mathbf{L}_r}, \mathbf{G}_r, \mathcal{T}), \tag{3}$$

under which we assume that the layout and glyph generation are independent of each other in our decoupled synthesis paradigm. Therefore, our objective is to model the layout distribution $P(\mathbf{L}_t | \mathbf{L}_r, \mathcal{T})$ and the glyph distribution $P(\mathbf{G}_t | \mathbf{G}_r, \mathcal{T})$, respectively.

**Facilitating Layout Generation with Offsets.** It is challenging to directly generate stylized handwriting with arbitrary layout structures and precise semantics through an end-to-end pipeline.

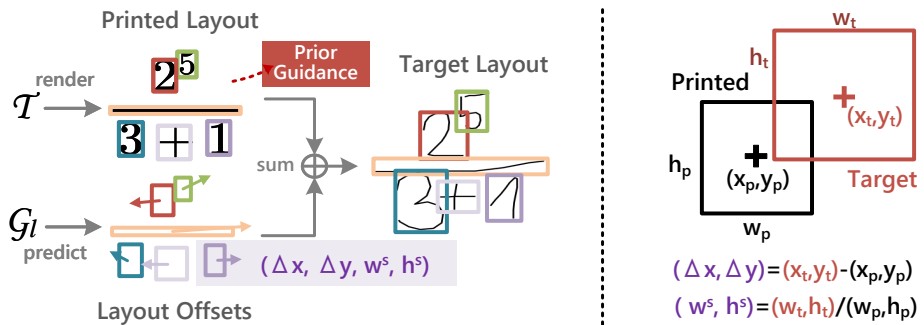

Figure 2: Concept of layout generation with offsets.

As shown in Fig. 2, to facilitate layout generation, we further render $\mathcal{T}$ into the printed template with the layout $\mathbf{L}_p$ and glyph embeddings $\mathbf{C}_p$, i.e., $\mathcal{T} \leftrightarrow (\mathbf{L}_p, \mathbf{C}_p)$, and then utilize the printed layout $\mathbf{L}_p$ as prior guidance, and finally infer the layout offsets $\Delta \mathbf{L} = (\mathbf{L}_t - \mathbf{L}_p)$ instead of absolute layout positions $\mathbf{L}_t$, under which the layout generation can be re-formulated as

$$P(\mathbf{L}_t | \mathbf{L}_r, \mathcal{T}) \Rightarrow P(\Delta \mathbf{L} | \mathbf{L}_r, \mathbf{L}_p, \mathbf{C}_p), \tag{4}$$

where $[\Delta x, \Delta y, w^s, h^s] \in \Delta \mathbf{L}$ denotes the offset (i.e., *the $x/y$ offsets of absolute positions and the scales of width and height*) of each bounding-box from $\mathbf{L}_p$ to $\mathbf{L}_t$. The layout generation becomes easier since the printed layout $\mathbf{L}_p$ provides strong prior knowledge (especially for OOV expressions) and the layout offsets $\Delta \mathbf{L}$ empirically are minor compared to the overall layout.

**Imitation through Implicit Context Adaptation (ICA)** To achieve stylized glyph-layout synthesis, we introduce implicit context adaptation (ICA) via cross-attention to jointly mimic layouts and calligraphic glyphs from reference examples. By treating reference layouts and glyphs as *external implicit contexts*, it selectively attends to relevant style features of each symbol and its bounding box. For instance, we leverage $\mathbf{G}_r$ as external contexts and perform ICA via cross-attention to mimic the calligraphic glyphs of reference samples, and thus $P(\Delta\mathbf{G}|\mathbf{G}_r, \mathcal{T}) \Rightarrow P(\mathbf{G}_t|\mathbf{G}_r, \mathbf{C}_p, \cancel{\mathbf{L}_p})$ can be modeled as

$$P(\mathbf{G}_t|\mathbf{G}_r, \mathbf{C}_p) \Rightarrow \xi\left(\mathrm{Softmax}\frac{\mathcal{Q}(\mathbf{C}_p) \times \mathcal{K}(\mathbf{G}_r)}{\sqrt{d}} \times \mathcal{V}(\mathbf{G}_r)\right), \tag{5}$$

where $\times$ denotes the matrix multiply, $\mathcal{Q}, \mathcal{K}, \mathcal{V}$ are projection operations, $\xi$ denotes the non-linear projection (such as multilayer perceptrons), and $d$ is the embedding dimension. Similarly, we can model $P(\mathbf{L}_t|\mathbf{L}_r, \mathcal{T})$ by feeding $\mathbf{L}_r$ as external contexts to obtain the final latent representations of target layouts.

## 3.2 NETWORK ARCHITECTURE

The overall framework for stylized HMEG is illustrated in Fig. 3, which follows a glyph-layout decoupled generation paradigm. Given the reference $\mathbf{X}_r \leftrightarrow (\mathbf{L}_r, \mathbf{G}_r)$ and textual content $\mathcal{T} \leftrightarrow (\mathbf{L}_p, \mathbf{C}_p)$, we aim to generate $\mathbf{X}_t \leftrightarrow (\mathbf{L}_t, \mathbf{G}_t)$ with similar structures and calligraphic styles of $\mathbf{X}_r$.

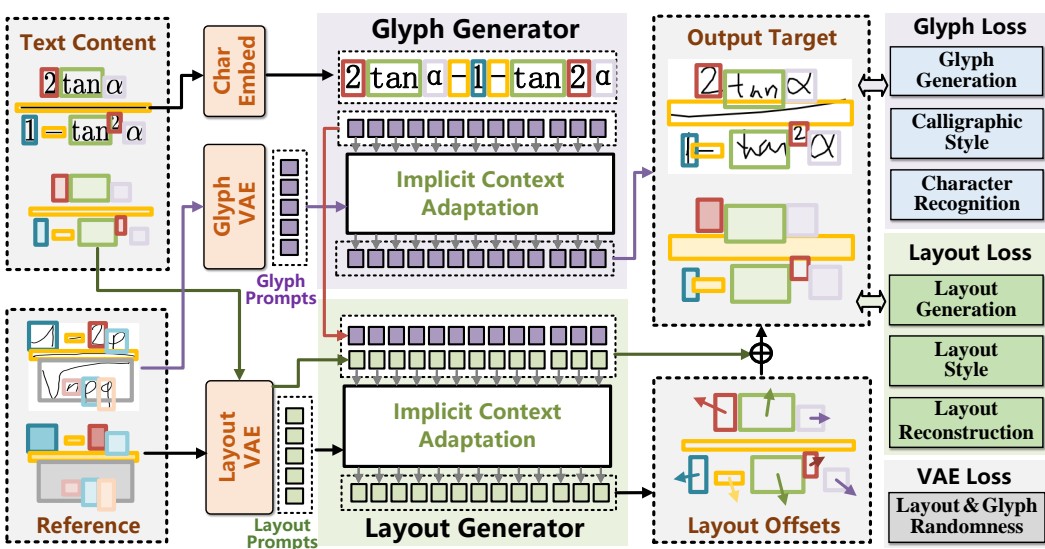

Figure 3: A layout-glyph decoupled framework for stylized HMEG.

**Glyph Generation via ICA** We leverage a variational auto-encoder (VAE) (Kingma & Welling, 2014) as the glyph encoder $\mathcal{E}_g$ to extract calligraphic features of handwriting as

$$\mathcal{E}_g(\mathbf{G}_r) = u_g + \exp(\frac{\sigma_g}{2}) \odot \epsilon, \tag{6}$$

where $u_g$ and $\sigma_g$ are the means and variances of $q_\phi(z|\mathbf{G}_r)$ that estimated by $\mathcal{E}_g$, and $\epsilon$ is a noise sampled from a standard normal distribution $p(z) \sim \mathcal{N}(0, 1)$. Then, we perform ICA (Implicit Context Adaptation) via cross-attention to obtain the target glyph embeddings as

$$\mathcal{F}_{\mathbf{G}} = \mathcal{I}\left(\mathcal{E}_g(\mathbf{G}_r), \mathbf{C}_p\right), \tag{7}$$

where $\mathcal{I}$ denotes the ICA via cross-attention as described in Eq. (5), and $\mathcal{F}_{\mathbf{G}} \in \mathbb{R}^{M \times d}$ are target glyph embeddings. We further decode each glyph embedding of $\mathcal{F}_{\mathbf{G}}$ into the sequences of point location probability (through Gaussian mixture models) and three pen states, denoted as $\left[\{\pi_r, \mu_r^x, \mu_r^y, \sigma_r^x, \sigma_r^y, \rho_r^{xy}\}_{r=1}^R, p^1, p^2, p^3\right]$ for each point, and we decode each sequence to the trajectory for presenting each glyph $g_m = \left\{[\Delta u_o, \Delta v_o, p_o^1, p_o^2, p_o^3]\right\}_{o=1}^O$ following Ha & Eck (2018) and obtain the target glyphs $\mathbf{G}_t = \{g_m\}_{m=1}^M$.

**Layout Generation via ICA** We also adopt the layout VAE $\mathcal{E}_l$ to extract latent presentations of layouts. Thus, target layout embeddings are calculated as

$$\mathcal{F}_{\mathbf{\Delta L}} = \mathcal{I}\left(\mathcal{E}_l(\mathbf{L}_r), \mathcal{E}_l(\mathbf{L}_p) \oplus \mathbf{C}_p\right), \tag{8}$$

where $\oplus$ denotes the element-wise addition. Then, we can directly regress $\mathcal{F}_{\mathbf{\Delta L}}$ to obtain the layout offsets as $\mathbf{\Delta L} = \{[\Delta x_l, \Delta y_l, w_l^s, h_l^s]\}_{l=1}^L$ and obtain the target layout as $\mathbf{L}_t = \mathbf{\Delta L} + \mathbf{L}_p$.

### 3.3 TRAINING OBJECTIVES

The entire framework is optimized with an end-to-end pipeline, where training objectives consist of glyph synthesis, layout synthesis, and VAE losses.

Specifically, *glyph synthesis objectives* include:

- **Glyph Generation Loss:** We maximize the likelihood of target glyphs $\mathbf{G}_t = \{g_m\}_{m=1}^M$ as

$$\mathcal{L}_g^{trj} = \mathbb{E}_{o,m}\left[-\log P([\Delta u_o, \Delta v_o, p_o^1, p_o^2, p_o^3]_m)\right] \tag{9}$$

  following Ha & Eck (2018), where each glyph $g_m$ contains $O$ trajectory points.

- **Character Recognition Loss:** The glyphs $\mathbf{G}_t = \{g_m\}_{m=1}^M$ are supposed to be recognizable, i.e.,

$$\mathcal{L}_g^{cls} = \mathbb{E}_m[-\tilde{c}_m \log \mathcal{R}(g_m)] \tag{10}$$

  where $\tilde{c}_m$ is the label of $g_m$, and $\mathcal{R}$ is a glyph recognizer.

- **Calligraphic Style Loss:** We further enforce the glyph encoder $\mathcal{E}_g$ to extract discriminative calligraphic styles. Concretely, we sample positive patch pairs $(p, p^+)$ from the same randomly selected glyph $g_m$ and construct negative pairs $(p, p^-)$ from other glyphs of different samples, and perform calligraphic contrastive learning as

$$\mathcal{L}_g^{sty} = \mathbb{E}_p\left[-\log \frac{\exp(s(p, p^+)/\tau)}{\sum_{p^-}\exp(s(p, p^-/\tau))}\right], \tag{11}$$

  where $s(p, p^+) = \delta(p)^\top \delta(p^+)$ denotes the similarity, $\delta$ denotes the linear projection, $p$ denotes the patch of a randomly selected glyph $g_m$, and $\tau$ is the temperature.

And, *layout synthesis objectives* include:

- **Layout Generation Loss:** We encourage the layout generator $\mathcal{G}_l$ to synthesize the target layouts $\mathbf{L}_t$ that are indistinguishable (by the discriminator $\mathcal{D}_l$) from the genuine layouts $\tilde{\mathbf{L}}$ as

$$\mathcal{L}_l^{adv} = \mathbb{E}_{\tilde{\mathbf{L}}}[\log \mathcal{D}_l(\tilde{\mathbf{L}})] + \mathbb{E}_{\mathbf{L}_t}\left[\log\left(1 - \mathcal{D}_l(\mathbf{L}_t)\right)\right], \tag{12}$$

  which follows a generative adversarial training paradigm.

- **Layout Reconstruction Loss:** layout consistency is improved via self-reconstruction constraint

$$\mathcal{L}_l^{rcn} = \mathbb{E}_{\mathbf{L}_r, \mathcal{T}_r}\left[||\mathbf{L}_r - \mathcal{G}_l(\mathbf{L}_r, \mathcal{T}_r)||_1\right], \tag{13}$$

  where $\mathcal{T}_r$ is the corresponding textual content of the reference $\mathbf{X}_r$, and $\mathcal{G}_l$ is the layout generator.

- **Layout Style Loss:** We further encourage the layout encoder $\mathcal{E}_l$ to learn precise style features of layouts. Specifically, we first sample positive patch pairs $(e, e^+)$ from the layouts of the same writer and prepare negative patch pairs $(e, e^-)$ from other layouts of different writers. Thus, we can achieve style constraints similar to Eq. (11) as

$$\mathcal{L}_l^{sty} = \mathbb{E}_e\left[-\log \frac{\exp(s(e, e^+)/\tau)}{\sum_{e^-}\exp(s(e, e^-/\tau))}\right]. \tag{14}$$

Lastly, **Layouts & Glyphs Randomness** is further introduced by incorporating *VAE objectives* as

$$\mathcal{L}_l^{vae} = \mathbb{E}_{\mathbf{L}_*}\left[D_{\text{KL}}(\mathcal{E}_l(\mathbf{L}_*)||\mathcal{N}(0,1))\right], \quad \mathcal{L}_g^{vae} = \mathbb{E}_{\mathbf{G}_r}\left[D_{\text{KL}}(\mathcal{E}_g(\mathbf{G}_r)||\mathcal{N}(0,1))\right], \tag{15}$$

where $D_{\text{KL}}$ is KL-Divergency, and the random noisy $\epsilon$ is sampled from normal distribution $\mathcal{N}(0,1)$.

Therefore, **overall training objectives** are formulated as

$$\mathcal{L}_{\mathcal{G},\mathcal{E}} = \underbrace{\mathcal{L}_g^{trj} + \lambda_g^{cls}\mathcal{L}_g^{cls} + \lambda_g^{sty}\mathcal{L}_g^{sty}}_{Glpyh} + \underbrace{\lambda_l^{adv}\mathcal{L}_l^{adv} + \lambda_l^{rcn}\mathcal{L}_l^{ren} + \lambda_l^{sty}\mathcal{L}_l^{sty}}_{Layout} + \underbrace{\lambda_g^{vae}\mathcal{L}_g^{vae} + \lambda_l^{vae}\mathcal{L}_l^{vae}}_{Randomness},$$

$$\tag{16}$$

where different $\lambda^*$ can be dynamically adjusted through the gradient balance as Fogel et al. (2020).

# 4 EXPERIMENTS

## 4.1 EXPERIMENTAL SETTINGS

**Datasets.** We leveraged CROHME (Mouchère et al., 2016) as the benchmark dataset, which comprises over 10K HMEs covering 126 categories of symbols with the bounding-box level annotations.

**Implementation Details.** All evaluations are conducted on a workstation with an RTX-4090 GPU. The whole model, implemented in PyTorch, is optimized via Adam (Kingma & Ba, 2014) with an initial learning rate of 0.0001 and reaches convergence with over 600K training iterations.

**Competitors.** Considering that stylized HMEG remains largely underexplored, we therefore only compare several approaches that can be adopted to this task, including **CycleGAN** (Zhu et al., 2017), **FormulaGAN** (Springstein et al., 2021), **Sg2Im** (Johnson et al., 2018), **SgHMEG** (Chen et al., 2024), and **ControlNet** (Zhang et al., 2023). *It is worth noting that conventional HTG methods (e.g., HiGAN+, VATr, GANwriting, etc.) are not applicable to HMEG, as they can only synthesize handwritten texts of linear structures.* In all comparisons, we use the official implementations with default settings or the publicly released pre-trained models (if provided) for a fair comparison.

**Evaluation Metrics.** We thoroughly evaluated different models regarding the following aspects:

- *Visual Quality:* **FID** and **SSIM** evaluate the visual quality of synthetic HMEs.
- *Semantic Correctness:* **ExpR**ate and **WER** measure the textual semantic correctness.
- *Glyph Similarity:* **HWD** (Pippi et al., 2023b) measures the calligraphic similarity of glyphs.
- *Layout Similarity:* We propose **LSD** (Layout Structure Distance; see Appendix A.3) to measure the layout similarity, which is accessed by rendering bounding boxes of layouts into mask images, and computing the FID score over those mask images, thereby measuring the distribution distance between the generated and real layouts. We also leverage **IOU** for reconstruction scenarios.

All evaluation networks designed for computing metrics (such as FID, WER, HWD, etc.) are independently pre-trained and are entirely separated from the generative models being optimized, and there is no overlap between their training data and ours, thereby eliminating potential data leakage.

**Evaluation Scenarios.** We thoroughly evaluated different models under the following scenarios:

- *Reconstruction.* It reconstructs all HMEs of the **training set** with identical texts and seen styles.
- *OOV Generation.* The model not only generates the stylized HMEs of OOV expressions and arbitrary structures, but it also mimics styles of unseen HMEs. Specifically, we evaluate under the OOV setting using the **official test set**, which guarantees no overlap with the training set, i.e., the writers are entirely distinct, and all test formulas (with their expressions and styles) are unseen.

## 4.2 COMPARISON WITH PREVIOUS SOTAS

| Method | Reconstruction | | | | | | | OOV Generation | | | | |
|---|---|---|---|---|---|---|---|---|---|---|---|---|
| | SSIM↑ | IOU↑ | FID↓ | WER↓ | ExpR↑ | HWD↓ | LSD↓ | FID↓ | WER↓ | ExpR↑ | HWD↓ | LSD↓ |
| **CycleGAN**[†] | 0.757 | × | 84.14 | 0.252 | 0.346 | 0.688 | × | 124.8 | 0.273 | 0.281 | 0.748 | × |
| **FormulaGAN**[†] | 0.724 | × | 74.68 | **0.134** | **0.451** | 0.717 | × | 141.9 | **0.112** | **0.519** | 0.753 | × |
| **ControlNet**[†] | 0.659 | × | 131.99 | 0.934 | 0.027 | 1.011 | × | 156.70 | 0.966 | 0.024 | 0.918 | × |
| Sg2Im | 0.787 | 0.324 | 10.02 | 0.287 | 0.235 | 0.452 | 8.158 | 30.35 | 0.447 | 0.162 | 0.473 | 7.919 |
| SgHMEG | 0.793 | 0.364 | 10.98 | 0.278 | 0.268 | 0.439 | 7.081 | 26.69 | 0.438 | 0.168 | 0.443 | 7.812 |
| **Ours** | **0.887** | **0.383** | **5.60** | 0.207 | 0.427 | **0.151** | **5.271** | **7.79** | 0.263 | 0.317 | **0.194** | 7.324 |

Table 2: Quantitative results of different models for stylized HMEG. †: the model cannot generate explicit layouts, and therefore cannot measure the layout similarity, such as IOU and LSD.

We first make a quantitative comparison of competing models under different evaluation setups, as shown in Table 2. It can be observed that our method signficantly outperforms previous SOTA models for HMEG in terms of visual quality, semantic and structural correctness, and style consistency.

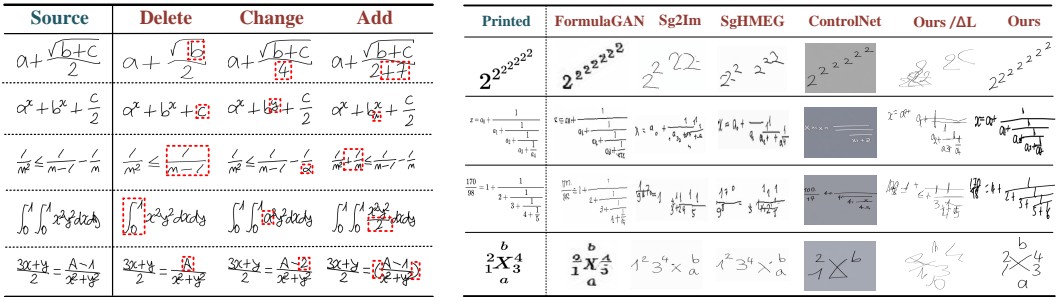

Figure 4: Qualitative results of different models from stylized HMEG.

We further provided a qualitative comparison in Fig. 4, and it can be observed that: **(1)** CycleGAN and FormulaGAN are more likely to produce printed-like HMEs, often generating unrealistic handwriting with low visual quality. **(2)** Sg2Im and SgHMEG typically produced blurred and distorted glyphs, and they also frequently generated incorrect structures of wrong semantics, especially for OOV generation. **(3)** Although ControlNet can generate realistic images, the textual semantics and layout structures are entirely incorrect. **(4)** Instead, our decoupled approach can generate realistic HMEs of precise layout structures and calligraphic styles in all scenarios.

## 4.3 GENERALIZATION ANALYSIS

Figure 5: Handwriting manipulation.    Figure 6: Generating HMEs of rare layout structures.

**Handwriting Manipulation in Text Space**    We demonstrate that our method can perform handwriting manipulation in text space, while strictly preserving the original structural and calligraphic styles. As shown in Fig. 5, given a source handwriting as the reference example, our model can edit given handwriting by adding, deleting, or changing its partial glyphs/symbols, whereas the edited HMEs preserve consistent layout structures and calligraphic styles of the source, demonstrating that our model can achieve effective stylized HMEG, rather than memorizing the training data.

**Generating Rare Layout Structures**    We further show a comparison of generating HMEs with rare layout structures in Fig. 6. We can observe that: FormulaGAN tends to produce unrealistic HMEs with blurred or distorted glyphs, ControlNet produces incorrect layouts and semantics, Sg2im and SgHMEG struggle to model complex structures. In contrast, our method via layout offsets can generate realistic HMEs of rare layout structures, demonstrating its strong generalization ability.

**Boosting HME Recognition via Generation** If a generative model can synthesize diverse realistic handwriting samples, it should be capable of improving recognition through generation-based augmentation. Therefore, we further validated whether our generative model can boost HME recognition (HMER), as shown in Table 3. We selected CAN (Li et al., 2022) as the HMER baseline and leveraged generative models to augment training data. It can be observed that our method successfully boosts HMER and outperforms the previous SOTA model for HMEG (i.e., SgHMEG).

| Method | Training Data | Datasets (ExpR↑) | | |
|---|---|---|---|---|
| | | C-14 | C-16 | C-19 |
| CAN (Base) | real | 0.464 | 0.506 | 0.542 |
| CAN + SgH. | real + *synth.* | 0.465 | 0.522 | 0.577 |
| CAN + Ours | real + *synth.* | **0.471** | **0.559** | **0.579** |

Table 3: Boosting recognition via generation.

| Method | Reconstruction | | | OOV Gen. | |
|---|---|---|---|---|---|
| | IOU↑ | ExpR↑ | LSD↓ | ExpR↑ | LSD↓ |
| Gaussian | 0.140 | 0.131 | 19.066 | 0.134 | 17.017 |
| Cond-RNN | 0.263 | 0.387 | 7.432 | 0.271 | 8.412 |
| Symbol Graph | 0.364 | 0.268 | 7.081 | 0.168 | 7.812 |
| Ours / $\Delta$L | 0.372 | 0.394 | 5.794 | 0.255 | 7.682 |
| Ours + $\Delta$L | **0.383** | **0.427** | **5.271** | **0.317** | **7.324** |

Table 4: Comparison results of different layout generation strategies.

### 4.4 ABLATION ON LAYOUT GENERATION STRATEGIES

As shown in Table 4, we further conducted an ablation study on different layout generation strategies, including: **Gaussian** (Yu et al., 2024), **Symbol Graph** (Chen et al., 2024), **Cond-RNN** (Ren et al., 2025), our cross-attention ICA w.o. layout offsets (i.e., **Ours / $\Delta$L**), and ours with layout offsets (i.e., **Ours + $\Delta$L**). Experiments show that our cross-attention ICA via layout offsets achieves the best results for layout generation, demonstrating its effectiveness for HMEG of arbitrary layouts.

### 4.5 HUMAN EVALUATION

**User Plausibility Study** We further conducted a human evaluation with 20 educated participants, who were shown random HMEs (half genuine and half generated) and asked to judge whether each was written by humans or machines, resulting in a total of 1200 responses for evaluation. As shown in Table 5, our model is perceived as plausible, with ∼50% of its outputs judged as human-written.

**User Preference Study** Participants were shown HMEs from each model with identical texts in random orders and asked to select the most preferred one. We repeated this procedure 40 times, resulting in 800 responses. As shown in Table 6, our model attains the majority of votes in all instances regarding overall layouts and quality, demonstrating its superiority over competing models.

| Actual Labels | Human Prediction | | Overall Accuracy |
|---|---|---|---|
| | Real | Fake | |
| Genuine | 0.276 | 0.224 | **0.513** |
| Generated | 0.263 | 0.237 | |

Table 5: User plausibility study.

| Target | C.GAN | F.GAN | C.Net | Sg2Im | SgH. | Ours |
|---|---|---|---|---|---|---|
| Layout[†] | × | × | × | 0.200 | 0.250 | **0.550** |
| HME | 0.069 | 0.150 | 0.036 | 0.150 | 0.164 | **0.431** |

Table 6: User preference study. †: Only methods that can generate explicit layouts are included.

### 5 CONCLUSION

Stylized handwriting of arbitrary structures and OOV expressions (such as stylized HMEG) remains largely underexplored, with existing methods falling short in addressing this challenge effectively. In this paper, we introduce a glyph-layout decoupled generation paradigm for stylized HMEG, which is more feasible than conventional I2I or G2I translation pipelines. Particularly, we propose to predict layout offsets from printed templates rather than generating absolute positions, where the prior guidance of templates simplifies the layout generation of arbitrary structures. Furthermore, we introduce implicit context adaptation (ICA) via cross-attention to jointly mimic layouts and calligraphic styles from reference examples. By treating reference layouts and glyphs as external implicit contexts, ICA selectively attends to relevant style features for each symbol and its bounding box. Experiments demonstrate that our method outperforms previous SoTA approaches for stylized HMEG.

ETHICS STATEMENT

This work advances stylized handwritten mathematical expression generation for applications in education, accessibility, and document digitization. All datasets used are publicly available, containing no personal or sensitive information. While the method could be misused (e.g., for academic dishonesty or forgery), our contribution is intended solely for research purposes, and we encourage responsible use. We acknowledge limitations in stylistic diversity and the environmental cost of training, and we have aimed to minimize computational overhead where possible.

REPRODUCIBILITY STATEMENT

This work has been conducted in accordance with standards for reproducibility in computational research. The conceptual outline, architecture details, training objectives, and hyperparameters for our proposed model are fully described in Section 3. All datasets used are publicly available and cited, with justifications for their selection. Comprehensive experimental details, including training configurations, evaluation scenarios, and evaluation metrics, are provided in Section 4.1.

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

# A APPENDIX

## THE USE OF LARGE LANGUAGE MODELS (LLMS)

We disclose that large language models (LLMs) were used solely to aid the writing of this paper, specifically for language polishing, including spelling and grammar checking. They did not contribute to the research ideation, methodology, experiments, analysis, or substantive writing.

## A.1 GENERATION RESULTS WITH DIFFERENT TEMPERATURES.

Despite relying on variational sampling, our method also enables the alternative controlled randomness during inference by applying temperature sampling to the output of the Gaussian Mixture Model, as illustrated in Fig. 7.

Figure 7: HMEG with various temperatures.

## A.2 LAYOUT VISUALIZATION OF DIFFERENT STRATEGIES.

To demonstrate the effectiveness of our layout generation strategies, we also provide a qualitative comparison of different methods in Fig. 8, including: **Gaussian** (Yu et al., 2024), **Symbol Graph** (Chen et al., 2024), **Cond-RNN** (Ren et al., 2025), our cross-attention ICA w.o. layout offsets (i.e., **Ours / ΔL**), and ours with layout offsets (i.e., **Ours + ΔL**).

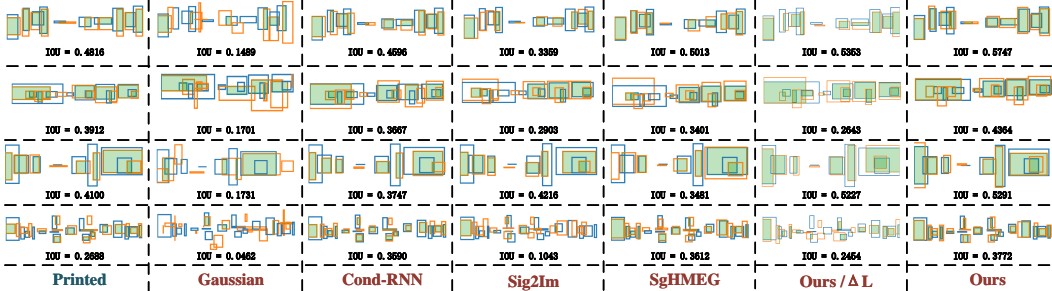

| Printed | Gaussian | Cond-RNN | Sig2Im | SgHMEG | Ours / ΔL | Ours |

Figure 8: Layout visualization of different strategies. Green areas indicate interactions between genuine and generated layouts.

## A.3 LAYOUT STRUCTURE DISTANCE (LSD).

To evaluate the layout similarity between generated and real layouts, we propose Layout Structure Distance (LSD). Given a layout, we convert it into a foreground–background mask image by rendering each bounding box as a filled foreground region on a blank background. This procedure discards the sequential ordering of boxes while preserving the overall spatial arrangement. The resulting mask images are passed through a pretrained Inception network to obtain high-level feature embeddings. Finally, we compute the Fréchet Inception Distance (FID) between the feature distributions of generated and real layouts, which serves as the LSD metric. While alternative implementations are possible, the overall idea and underlying principle of LSD remain consistent with our approach.