# OpenReview forum: "Stylized Handwriting Generation of Arbitrary Structures and OOV Expressions: A Decoupled Approach via Layout-Offsets"
_ICLR.cc/2026/Conference — ICLR 2026 Conference Withdrawn Submission_

### Official Review · Reviewer_9KiC · 2025-10-27

**Soundness:** 1
**Presentation:** 3
**Contribution:** 2
**Rating:** 2
**Confidence:** 5

**Summary:**

This paper focuses on the generation of stylized handwriting with 2D structure, focusing on Handwritten Mathematical Expressions). The authors identify 2D spatial layout, style coherence, and out-of-vocabulary generalization as the main difficulties for he task. To address this, they propose a 2-stage model which first generates the layout, followed by writing each individual symbol relative to the appropriate bounding box in the generated layout. Authors propose an approach they call "Implicit Context Adaptation" which is a cross-attention mechanism to incorporate the layout from the printed text expression, and the style from the handwriting, into the generation of the text content in the provided style. The decoder conditioned on the extracted style follows sketch-rnn / "Generation of sequences with recurrent neural networks" from Graves et al, modelling the output distribution using a mixture of Gaussians for offsets + pen-up Bernoulli. Authors compare their approach to image-generation-based approaches (noting that handwriting-generation-based approaches would not work for the two-dimensional structure of handwritten mathematics), namely CycleGAN, FormulaGAN, Sg2Im, SgHMEG, and ControlNet, on the CROHME'16 test set, measuring image-based metrics (FID, SSIM), handwriting-recognition-based metrics (WER, ExpRate), as well as metrics to measure handwriting similarity and layout similarity. Test set is used to evaluate for OOV expressions (the writers and expressions in the test set are unique). Qualitative study (human eval for real vs synthetic and comparison with other approaches) is also performed.

**Strengths:**

- Originality: The core idea is somewhat novel and interesting, and should allow to generate quite complex structures, as long as the layout / segmentation for such structures can be defined. The proposed "Implicit Context Adaptation" idea does not carry novelty as (evident from Eq. 5) it is a fairly standard cross-attention mechanism but that does not take away from the idea of two-stage generation.
- Quality & Clarity: The writing of the paper is fairly clear and is likely easy to reproduce. The comparison with the proposed approaches is quite detailed, with several quantitative metrics and a qualitative study. However, there are a number of concerns regarding limited evaluation, ablation study, and comparison with other works - see 'Weaknesses' below.
- Significance: The paper would be quite significant if the results were to show a clear and undeniable advantage of the proposed approach compared to others. Reliance on printed layout limits applicability to cases where a pre-rendered template is available. The expressiveness of style transfer is also limited (ex. if a human writes a fraction with a diagonal bar '/' but the user template always uses the horizontal bar '-', this would not be useable)

**Weaknesses:**

The 4 main weaknesses of the paper all have to do with the evaluation:
1. All of the methods which the proposed method compares to are image-generation-based, basically making this 'apples-to-oranges' comparison because these approaches exhibit different biases (in particular, the qualitative study of 'whether real or synthetic' in table 5 would be subject to background artifacts for one but not the other). While authors note 'It is worth noting that conventional HTG methods
(e.g., HiGAN+, VATr, GANwriting, etc.) are not applicable to HMEG, as they can only synthesize handwritten texts of linear structures', the authors provide no concrete proof of this fact, by comparing to sketch-rnn-like approaches (which are able to generate spatial structures like sketches). The paper would be strengthened by doing such comparisons.
2. The evaluation is performed only on 1 dataset (CROHME'16), while ignoring newer datasets, with more complex structures (such as CROHME'19, CROHME'23, or MathWriting, all of which contain some sort of character segmentation information - which is also only needed for training, but not for evaluation (with the exception of the LSD metric, which is proposed by the authors)). The paper would be significantly strengthened by a larger evaluation.
3. The results from Table 2 show that handwriting generated by the proposed approach is 50%/136% less recognizable than what is generated by the FormulaGAN approaches (as measured by WER metric). The paper would be strengthened by providing an explanation of this fact (which is not given in lines 399-404 discussing the table).
4. The ablation study compares several versions of the layout generation but does not measure the performance of the system completely without the layout generation models. Similarly, no study is done on the effect of 7 different losses introduced by the authors.

**Questions:**

Could you please provide details on:
1. The performance gap with FormulaGAN
2. The details of handwriting editing (lines 421-426). Is the whole ink being regenerated, or only new symbols? The results in Figure 5 suggest that regeneration could affect the scale of the ink (ex. line 3) - or is that an artifact or needing to render a longer ink in the same space?
3. Could you please comment on the performance of your proposed system in more complex layout structures (ex. matrices?). Most of the expressions in Figures 7, 5, 6, and 4 are still fairly linear in their nature.

---

### Official Review · Reviewer_P34U · 2025-10-30

**Soundness:** 2
**Presentation:** 3
**Contribution:** 2
**Rating:** 4
**Confidence:** 5

**Summary:**

This paper aims at stylized handwritten mathematical expression generation. It proposes a glyph-layout decoupled method, achieving the generation of stylized handwriting of arbitrary layout structures by first synthesizing layout offsets and then generating stylized glyphs. Experiments evaluate the proposed method.

**Strengths:**

1.	This paper provides a very detailed description of the proposed method in Section 3 (Methodology), which aids in understanding the method.
2.	This paper thoroughly compares different methods under two scenarios: "Reconstruction" and "OOV (Out-of-Vocabulary) generation".

**Weaknesses:**

1. Please clarify the practical applications of stylized handwritten mathematical expression generation(HMEG). The greater value of HMEG should be in generating data with random styles to augment the recognition dataset, rather than achieving controllable styles.

2. The idea of glyph-layout decoupled generation is not novel; a similar idea is proposed in [1].

3. The proposed implicit context adaptation (ICA) using cross-attention, where the text content serves as the query to retrieve information from reference examples. It is common in handwritten text generation [2][3] and not novel.

4. There are other methods[2][4][5] for generating handwritten text; please provide a more comprehensive discussion in Section 2.1.

5. Figure 4 lacks the ground truth or reference examples; which make it impossible to qualitatively compare the ability of different methods to learn calligraphic styles.

6. The ablation study is insufficient.

&emsp;&emsp; The comparison with the joint generation of glyphs and layouts is lacking;

&emsp;&emsp; The ablation of the proposed prior guidance of template layouts is lacking;

&emsp;&emsp; The ablation of ICA in glyph generation is lacking.

[1] Decoupling layout from glyph in online chinese handwriting generation, ICLR 2025.

[2] Handwriting Transformers, ICCV 2021.

[3] Handwritten Text Generation from Visual Archetypes, CVPR 2023.

[4] JokerGAN: memory-efficient model for handwritten text generation with text line awareness,ACM MM 2021.

[5] One-DM:One-Shot Diffusion Mimicker for Handwritten Text Generation, ECCV 2024

**Questions:**

1. What are the application values of stylized handwritten mathematical expression generation?
2. It is recommended to explain the novelty of the proposed glyph-layout decoupled generation and ICA.
3. It is recommended to supplement necessary ablation studies; for details, refer to the "weakness".
4. During the inference process, how many reference examples are required for each generation?

---

### Official Review · Reviewer_WaoU · 2025-10-31

**Soundness:** 2
**Presentation:** 2
**Contribution:** 2
**Rating:** 4
**Confidence:** 5

**Summary:**

This paper focuses on stylized generation of handwritten mathematical expressions with arbitrary layouts and out-of-vocabulary symbols. To this end, it proposes a glyph–layout decoupled framework that predicts layout offsets from printed templates instead of absolute positions, and employs cross-attention-based implicit context adaptation to jointly mimic reference glyphs and spatial styles. Extensive experiments on CROHME show that the approach surpasses prior SOTA in visual quality, structural/semantic correctness, and style consistency, while also boosting downstream expression recognition.

**Strengths:**

1. The strong prior of printed templates enables robust handling of Out-of-Vocabulary (OOV) expressions and accurate style mimicry.
2. This paper provides thorough experiments in different evaluation scenarios, human judgments, and downstream recognition gains.

**Weaknesses:**

1.	The core decoupling idea closely resembles that in [1], which diminishes the innovativeness of the method.
2.	Qualitative experiments lack style references. Please provide visual pairs that align the reference expression with the corresponding generated expressions.
3.	Template-offset prediction relies on strong prior information about the printed template. Perhaps some robustness experiments could be provided, such as when there is a significant deviation between the reference style and the printed template, to analyze the applicability of this offset prediction method.
4.	The paper does not include any ablation studies to isolate the contribution of individual components.

[1] Decoupling layout from glyph in online Chinese handwriting generation, ICLR 2025

**Questions:**

Please refer to Weaknesses 1-4.

---

### Official Review · Reviewer_5AXy · 2025-11-01

**Soundness:** 3
**Presentation:** 3
**Contribution:** 3
**Rating:** 6
**Confidence:** 4

**Summary:**

This paper presents a significant advancement in handwriting generation, pioneering stylized handwriting with arbitrary layout structures. This highly novel contribution addresses a critical gap in the field.

**Strengths:**

1. A novel dual-branch framework for HMEG, inspired by human handwriting, effectively decouples layout and glyph generation to achieve SOTA performance.
2. The introduction of offset generation significantly enhances the model's ability to discern stylistic differences between machine-printed and handwritten elements, leading to superior style transfer.
3. Implicit Context Adaptation, coupled with separate loss functions, effectively captures and mimics stylistic discrepancies, ensuring both high imitation fidelity and stylistic variability.

**Weaknesses:**

1. While introducing stylized HMEG as a novel task focusing on stylized generation with arbitrary layouts, the paper lacks a sufficiently detailed discussion on its practical significance, especially when contrasted with the more readily apparent utility of tasks such as arbitrary layout recognition or structured (e.g., paragraph-level) stylized text generation.
2. The experimental evaluation section suffers from a lack of clarity regarding the setup of comparative experiments. Specific details are missing on how the proposed method performs reconstruction (e.g., whether reference and prediction originate from the same image). Furthermore, the paper does not adequately describe how existing methods handle Out-Of-Vocabulary (OOV) generation or mimic styles from unseen HMEs in these comparative evaluation scenarios, which hinders a thorough understanding and fair assessment of the results.

**Questions:**

Given the novelty of stylized HMEG, I'm concerned about the direct comparability of baselines in Table 2. Baselines (e.g., I2I/G2I) likely memorize printed styles. In contrast, your method explicitly uses a reference style character, introducing a stronger prior that baselines may lack. Please clarify how this impacts the fairness of comparison.

---

### Note · Authors · 2025-11-13

**Comment:**

We sincerely thank the reviewers for their valuable time and constructive feedback. After careful consideration, we have decided to withdraw the paper at this stage and will further improve it based on the comments.

**Withdrawal Confirmation:**

I have read and agree with the venue's withdrawal policy on behalf of myself and my co-authors.